# Genome-Wide Identification and Expression Analysis of the RADIALIS-like Gene Family in *Camellia sinensis*

**DOI:** 10.3390/plants12173039

**Published:** 2023-08-24

**Authors:** Shaoying Wang, Beibei Wen, Yun Yang, Shanshan Long, Jianjun Liu, Meifeng Li

**Affiliations:** College of Tea Sciences, Guizhou University, Guiyang 550025, China; w18224999471@126.com (S.W.); bbwen@gzu.edu.cn (B.W.); yy17585113068@163.com (Y.Y.); 15121686642@163.com (S.L.)

**Keywords:** RADIALIS-like (RL), bioinformatics analysis, abiotic stress, self-activation, yeast two-hybrid system

## Abstract

The RADIALIS-like (RL) proteins are v-myb avian myeloblastosis viral oncogene homolog (MYB)-related transcription factors (TFs), and are involved in many biological processes, including metabolism, development, and response to biotic and abiotic stresses. However, the studies on the *RL* genes of *Camellia sinensis* are not comprehensive enough. Therefore, we undertook this study and identified eight *CsaRLs* based on the typical conserved domain SANT Associated domain (SANT) of RL. These genes have low molecular weights and theoretical p*I* values ranging from 5.67 to 9.76. Gene structure analysis revealed that six *CsaRL* genes comprise two exons and one intron, while the other two contain a single exon encompassing motifs 1 and 2, and part of motif 3. The phylogenetic analysis divided one hundred and fifty-eight RL proteins into five primary classes, in which CsaRLs clustered in Group V and were homologous with CssRLs of the *Shuchazao* variety. In addition, we selected different tissue parts to analyze the expression profile of *CsaRLs*, and the results show that almost all genes displayed variable expression levels across tissues, with *CsaRL1a* relatively abundant in all tissues. qRT-PCR (real-time fluorescence quantitative PCR) was used to detect the relative expression levels of the *CsaRL* genes under various abiotic stimuli, and it was found that *CsaRL1a* expression levels were substantially higher than other genes, with abscisic acid (ABA) causing the highest expression. The self-activation assay with yeast two-hybrid system showed that CsaRL1a has no transcriptional activity. According to protein functional interaction networks, CsaRL1a was well connected with WIN1-like, lysine histidine transporter-1-like, β-amylase 3 chloroplastic-like, carbonic anhydrase-2-like (CA2), and carbonic anhydrase dnaJC76 (DJC76). This study adds to our understanding of the RL family and lays the groundwork for further research into the function and regulatory mechanisms of the *CsaRLs* gene family in *Camellia sinensis*.

## 1. Introduction

Environmental stress adversely affects plant growth, development, and metabolite content. Plants have developed coping mechanisms against harsh environments, including molecular and metabolic responses, as part of their adaptive response. Plant transcription factors (TFs) are the key molecular targets in modulating passive mechanisms [1,2]. The MYB superfamily has the most TF members in the plant kingdom and plays a central role in abiotic stress signaling and secondary metabolism regulation [3,4,5]. The RADIALIS-like (RL) family is a subgroup of I-box-binding-like MYB-related TFs with a highly conserved SANT domain [6,7,8,9], which has been linked to a variety of physical and biochemical processes such as embryonic development [10], morphological traits [11], pigmentation [12], and ABA signaling [13].

The RL transcription factors are essential for plant physiological and reproductive growth. The RL proteins are highly homologous to the RADIALIS (RAD) protein of *Antirrhinum*, establishing dorsal-ventral asymmetry [14]. The over-expression of *FSM1* (a Radialis-like gene) explicitly expressed in the early stages of tomato fruit development, resulted in narrower cotyledons, slower growth, reduced apical dominance, and even a deformed appearance of the young tomato plants [11]. SlFSM1 induces developmental changes via the FSM1-FSB1 complex, negatively affecting cells with the greatest potential for expansion [15]. In *Arabidopsis*, AtRL2 (AtRSM1 or AtMEE3) interacts with the *HOOKLESS 1* (*HLS1*) gene, and over-expression of *AtRL2* results in a lack of apical hooks with short hypocotyls, a defect in gravitropism during early morphogenesis, and hypersensitivity to red light during early photomorphogenesis [16]. Additionally, AtRL2 targets the FLC promoter to increase FLOWERING LOCUS C (*AtFLC*) expression while suppressing *FLOWERING LOCUS T* (*AtFT*), which results in repressed floral transition [17]. *GbRL1*, a member of the RL family in *Gossypium barbadense*, may have a role in fiber differentiation [18]. The RNA interference of *PhRL* expression in *Petunia* resulted in a significant decrease in pollen viability and the enlargement of sexual organ size, indicating that *PhRL* is a fertility-related gene and may be involved in both anther and pistil development [19]. *PvRL1* and *PvRL2* of *Plukenetia volubilis* were involved in flower sex determination [20].

The RL proteins are also involved in hormone signaling and the response to abiotic stress. OsRL3, for example, promotes dark-induced leaf senescence and reduces susceptibility to salt stress in rice via the abscisic acid (ABA) signaling pathway [13]. AtRL2 is involved in HLS1-mediated auxin signaling during early photomorphogenesis in *Arabidopsis* [16]. Furthermore, AtRL2 promotes ABA Insensitive 5 *(AtABI5*) expression by converging on its promoter and improving plant adaptation to harsh environments [21]. Many other *RLs* respond to abiotic stresses such as drought, elevated carbon dioxide, excessive light, nitrogen stress, salinity, and piezoelectric ultrasound [22,23,24,25,26,27].

Secondary metabolism is also closely linked to the RL proteins. CsaRL1a, for example, was predicted to be involved in the biosynthesis of glycosyl flavonoids in *Camellia sinensis* [22]. *AcRL4* is linked to striped leaf albinism in *Areca catechu* L. [28]. *TfRAD1* over-expression disrupted the asymmetric corolla pigmentation pattern in *Torenia fournieri* by significantly downregulating anthocyanin biosynthesis genes [12]. The heterotrimeric complex PtRAD1 (PtRL3)-PtDRIF1-PtWOX13c promotes xylem formation in *Populus*, whereas PtRAD1-PtDRIF1-PtKNAT7 inhibits secondary cell wall formation in xylem [29,30].

Tea is one of the most popular drinks in the world due to its economic value and health benefits. Although the evolution and diversity of RL families have been studied in many flowering plants, including *Arabidopsis*, *Dipsacales*, *Plantago*, and *Oleaceae* [14,31,32,33,34], there has been no comprehensive study of the *RL* family genes in tea to date. Using the genome of *Camellia sinensis* var. *assamica* cv. *Yunkang-10* [35], we investigated the putative functions of *RL* family genes in tea. The *CsaRL* family genes were cloned and characterized by investigating their structure, protein properties, motifs, conserved domains, and phylogenetic relationship. The tissue-specific profile and stress responses of *CsaRLs* were detected using quantitative real-time PCR (qRT-PCR). We also visualized the CsaRL1a protein–protein interaction network and predicted the target proteins. Our findings will aid future research into the functional identification and regulatory mechanisms of *RL* genes in tea.

## 2. Results

### 2.1. Identification and Cloning of CsaRLs in the Yunkang-10 Variety of Tea

The RL proteins of *Arabidopsis*, *Oryza sativa*, and *Solanum lycopersicum* were utilized as queries to identify and obtain the *CsaRL* genes in the *Yunkang-10* tea plants using BLAST. Furthermore, we employed the SANT domain (PF00249) to carry out HMM searches against the local protein database with HMMER3.0. Further domain analysis and the elimination of duplicated sequences revealed the presence of eight CsaRL proteins in the tea genome. These *CsaRL* genes were designated as *CsaRL1a*, *CsaRL1b*, *CsaRL3a*, *CsaRL3b*, *CsaRL3c*, *CsaRL4a*, *CsaRL4b*, and *CsaRL4c*, according to their homology to the *CssRL* genes in the *Shuchazao* (tpdb.shengxin.ren) variety, based on the NCBI description. The mRNA was extracted from tea plant tissues and reverse transcribed into single-strand cDNA, and the *CsaRL* family genes in *Yunkang-10* were cloned using specific primers (Appendix A). As shown in Table 1, the percent identity of *CsaRLs* with *CssRLs* ranges from 89.24% to 99.67%, indicating the conservation of *RL* family genes in tea plants. *Csa4a* and *Csa4b* are found at separate chromosomal locations in the genome of *Yunkang-10* but share an identical nucleotide sequence and 99.67% identity with *CssRL4* from cv. *Shuchazao*.

Table 2 shows the bio-information for CsaRL proteins, which includes the protein name, amino acid length, molecular weight (kDa), isoelectric point (p*I*), GRAVY, and predicted location. Csa4a and Csa4b have identical protein sequences and the same physical and chemical parameters due to their similar nucleotide sequences. All CsaRLs have small molecular weights ranging from 8.80 kDa (CsaRL1a) to 11.38 kDa (CsaRL4a and CsaRL4b). The isoelectric points (p*I*s) of the CsaRLs vary from 5.67 (CsaRL3b) to 9.76 (CsaRL3c), and the aliphatic index value ranges from 49.31 (CsaRL4a and CsaRL4b) to 73.51 (CsaRL3c). The GRAVY value of all CsaRLs is less than zero, indicating that CsaRLs are hydrophilic. CsaRL proteins are all predicted to be located in the nucleus.

### 2.2. Analysis of Gene Structure, Protein Motifs, and Conserved Domains of CsaRLs

We examined the gene structure, protein motifs, and conserved domains to investigate the diversification of *CsaRL* genes. Six *CsaRLs* had two exons and one intron, whereas *CsaRL1b* and *CsaRL3a* only had one exon (Figure 1A). Among the eight CsaRL protein sequences, five motifs were detected, with motifs 1–3 shared by all sequences. CsaRL4a and CsaRL4b differed significantly from other members in the C-terminal regions due to the extra motif 5 (Figure 1B). Sequence alignment showed the CsaRL proteins had relatively similar lengths, varying from 77 to 101 amino acids, with all eight CsaRLs having remarkably conserved SANT domains in the N-terminal and putatively incomplete CREB 2 domain in the C-terminal (11) (Figure 1C).

### 2.3. Phylogenetic Analysis

To investigate the evolutionary relationship and functional divergence of CsaRL proteins, the full-length amino acid sequences of 158 RL proteins from 18 plant species were downloaded and used to construct a phylogenetic tree using the maximum likelihood technique in MEGA7 (Figure 2). All of the RL members were divided into five groups. The RLs of dicots and monocots were included in Group I, III, IV, and V, while Group II solely included *Oryza* monocots.

The Group V is the most diverged and contains RLs from monocotyledons, dicotyledons, herbs, forests, aquatic plants, and terrestrial plants. Previously studied RLs are highlighted in bold, such as AtRL1–4 (NP 195636.2, NP 179759.1, NP 001077912.1, and XP 002869031.1) of *Arabidopsis*, SlFSM1 (XP 001233849.1) of *Solanum lycopersicum*, OsRL3 (XP 015627089.1) of *Oryza sativa*, and PtRAD1 of *Populus*. The CsaRLs and CssRLs, highlighted in red and green, were clustered together in Group V. The distribution of CsaRLs in the tree is related to their motif distribution. CsaRL1a and CsaRL1b, sharing the same motif distribution, were present in subgroup 1 of Group V and were homologous with CssRL1 (XP 028084596.1). CsaRL3b was distant from other CsaRLs due to the loss of motif five and was homologous with CssRL3 (XP 028058740.1). Other CsaRLs with gaps between motif three and motif five were distributed in subgroup 4 of Group V and were highly homologous with CssRL3 isoforms (XP 028058086.1, XP 028054259.1, XP 028054260.1, and XP 028054262.1, respectively) and CssRL4 isoforms (XP 028062160.1 and XP 028082142.1).

### 2.4. Tissue-Specific Profiling of CsaRL Family Genes in Tea

To predict the roles of *CsaRL* genes in tea, 12 separate tissue samples from *Yunkang-10* were harvested and assessed for their expression pattern using quantitative real-time PCR, including flowers, apical buds, 1–5 leaves, mature leaves, upper tender stems, middle tender stems, lower tender stems, and roots. Since *CsaRL4a* and *CsaRL4b* have identical nucleotide sequences, it is challenging to distinguish between their expressions. As a result, we chose *CsaRL4a* and *CsaRL4b* to represent their combined expression. The profiling pattern was divided into two groups based on expression abundance. *CsaRL1b*, *CsaRL4a*, and *CsaRL4b* were part of the first group and had lower expression in almost all tissues. The other genes belonged to the second group and had relatively higher expression in all tissues (Figure 3). *CsaRLs* were expressed differently in each of the 12 tissues studied. *CsaRL1b*, *CsaRL3a*, *CsaRL3c*, *CsaRL4a*, and *CsaRL4b* were expressed at minimal levels in all tissues except the lower stems, suggesting that these genes may be involved in xylem development. *CsaRL3b* and *CsaRL4c* were more highly expressed in roots, fourth leaves, fifth leaves, and lower stems, suggesting that they may play essential roles in roots, xylem, and leaves during the maturation period. *CsaRL1a* was significantly downregulated in the young tissues of flowers, buds, and upper stems, but significantly upregulated in developing and mature tissues, especially from second to mature leaves. Moreover, in the fourth and fifth leaves, *CsaRL1a* was expressed substantially more than other genes, up to 17-fold and 32-fold more than *CsaRL4c*, the second-highest expressed gene. These findings suggest that *CsaRL1a* is the most crucial functional gene in the *CsaRL* gene family in tea.

### 2.5. Expression Patterns of CsaRL Genes under Different Abiotic Stresses

Tea seedlings were harvested after 72 h of abiotic stress treatments, including drought (PEG-6000), salinity (NaCl), methyl jasmonate (MeJA), and abscisic acid (ABA), to investigate the involvement of the *CsaRL* genes in response to different abiotic stresses. The *RL* family genes responded differently to each stress treatment (Figure 4). Almost all *CsaRLs* genes could be induced in plants after 72 h of exposure to PEG-6000, NaCl, MeJA, and ABA. The expression patterns of the upregulated genes tended to increase first and then decline. Under the PEG-6000 treatment, we observed an increased expression of multiple *CsaRLs*. Among them, the expression levels of *CsaRL1a/1b/3a/3c/4a&4b/4c* increased by more than two-fold at least at one of the time points. The expression levels of *CsaRL1a* dramatically increased throughout the PEG-6000 treatment period. *CsaRL1a* displayed the highest expression levels after 48 h of PEG-6000, which were more than 27-fold higher than those at 0 h of PEG-6000 (Appendix A). Other genes, such as *CsaRL3a/3b/3c*, likewise demonstrated the highest levels of expression after 48 h of PEG-6000 treatment. PEG-6000 did not induce *CsaRL1b* at 12 h, 24 h or 48 h or *CsaRL3b* at 12 h, 24 h, or 72 h. NaCl stress triggered many *CsaRLs* up to different degrees: *CsaRL1a*, *CsaRL3b*, *CsaRL3c*, *CsaRL4a*, and *CsaRL4b* significantly increased after 48 h of NaCl treatments. *CsaRL1b* and *CsaRL3a*, on the other hand, showed the highest levels of expression at 24 h of stress. Their expression levels were much greater than the control, indicating that these genes could play a vital role in avoiding damage to the tea plant caused by NaCl-induced stress conditions. MeJA stress activates almost all *CsaRL* genes. *CsaRL1a*, *CsaRL1b*, *CsaRL3a*, *CsaRL3b*, *CsaRL3c*, *CsaRL4a*, and *CsaRL4b* exhibited a considerable increase in expression levels at 48 h. However, in contrast to PEG-6000, NaCl, and MeJA treatments, most *CsaRLs* attained peak expression levels after 24 h of ABA treatment. *CsaRL1a* increased steadily until 24 h of ABA. At the 24 h time point of ABA treatment, *CsaRL1* expression increased more than 87-fold relative to the control, which was substantially greater than the expression levels of other genes (Figure 4). Interestingly, the expression patterns of most *CsaRL* genes, excluding *CsaRL1b* and *CsaRL4c*, under PEG-6000 treatment showed a trend of first increasing and then decreasing. In addition, almost all *CsaRL* genes were induced when exposed to prolonged treatment with NaCl, MeJA, and ABA. In tea plants, ABA treatment could transiently stimulate the expression of several *CsaRL* genes in a short period. Notably, under various treatments, the expression level of *CsaRL1a* grew much more than that of the others and was strongly triggered by ABA. The stress responses reveal that *CsaRL1a* is critical for plant resilience, particularly in the presence of ABA. In *Arabidopsis*, downregulation or upregulation of *RSM1* expression changes the sensitivity of seed germination and cotyledon greening to ABA and NaCl [21]. Overall, drought, salinity, methyl jasmonate, and abscisic acid treatment lowered the expression patterns of the *CsaRL* genes in tea plants.

### 2.6. Transcriptional Activation Activity of CsaRLs

Most RL proteins were annotated as TFs, so we analyzed the transcriptional activation. We employed the GAL4-responsive reporter system in yeast to measure the transactivation activity of CsaRL proteins. The transcriptional activation activity of CsaRLs was analyzed using a yeast assay. The empty vector pGBKT7 and a recombinant plasmid containing pGBKT7-CsaRLs and pGBKT7-GAL4 DNA-binding domain were transformed into the yeast strain Y2HGold, which has the reporter gene TRP1 plus upstream activating sequences. When the reporter gene TRP1 is activated, it indicates the yeast with the plasmid can grow on media without histidine (SD-trp). We used the yeast expression vector pGBKT7 as the negative control and the plasmid pGBKT7-GAL4 as the positive control. The transformed yeast cells were selected on SD/-Trp media and then cultured on SD/-Trp or SD/-Trp/-His/-Ade media. The results revealed that all yeast transformants grew well on SD/-Trp medium. However, only the yeast transformant containing pGBKT7-GAL4 showed the blue pigment and survived on SD/-Trp/-His/-Ade media, the yeasts with CsaRL plasmids did not grow on the same medium and had no transactivation activity (Figure 5). These results indicated that the CsaRLs have no transcriptional self-activation activity.

### 2.7. Functional Interaction Networks of CsaRL1a Based on Transcriptome Data

Based on the tissue-specific profiling and stress-dependent expression of *CsaRL* family genes, *CsaRL1a* is assumed to be involved in the growth, development, and abiotic stress responses. Using the web software Lianchuan Omicstudio, we mapped the functional interaction networks of CsaRL1a based on transcriptome data [22] (Figure 6). The network showed that CsaRL1a is closely correlated with an ethylene-responsive transcription factor, WIN1-like, auxin efflux carrier, pectate lyase, and chaperone proteins. Most of these proteins are involved in stress response, transportation, hormone signaling, and primary and secondary metabolism. Pectate lyase and chaperone proteins mediate stress reactions, whereas lysine hydroxymethyltransferase, UFGT3-like transferase, and GDSL esterase/lipase are strongly associated with stress response and metabolism. This result implies that CsaRL1a also plays a role in regulating metabolic biosynthesis, in addition to roles in growth, development, and stress tolerance.

## 3. Discussion

The MYB family of transcription factors is the largest and most functionally diverse in the plant kingdom [36]. The RL proteins are classified as I-box-binding like MYBs [9]. To date, the evolution and function of the RL gene family in numerous flowering plants, including *Arabidopsis*, *Antirrhinum*, *Torenia fournieri*, *Orchis italica*, and *Oryza* [12,13,14,31], have been studied. However, there has been no extensive study of *RL* genes in tea.

### 3.1. The Unique Sequence Characteristics of CsaRL Proteins Are Related to Molecular Functions

In this study, we cloned and characterized eight *CsaRLs* from *C. sinensis* var. *assamica* cv. *Yunkang-10*. The percent identities of *RL* family genes in cv. *Yunkang-10* and cv. *Shuchazao* varied from 89.24% to 99.67%, indicating that *RL* genes in tea plants are homologous and have tight evolutionary relationships [37]. CsaRLs have molecular weights ranging from 8.80 Kd to 11.38 Kd, showing that RL proteins have small tertiary structures. The tertiary structure of proteins is responsible for molecular interactions, including protein–protein and protein–nucleic acid interactions, and it is involved in many essential biological processes, such as signal transduction, transport, cell regulation, gene expression control, enzyme inhibition, antibody-antigen recognition, and multi-domain protein assembly [38]. The small tertiary structure of RLs facilitates the formation of protein complexes and binding to target gene promoters. In *Populus*, a heterotrimer PtrRL1-PtDRIF1-PtWOX13c, formed possibly due to PtDRIF1′s inability to accept larger PtDIV proteins, is involved in multiple interactions that modulate an array of processes [30]. In molecular interactions, the isoelectric point (p*I*) is also crucial [39]. The p*I*s of eight CsaRLs range from 5.67 to 9.76, indicating that the optimal conditions for CsaRLs in protein–protein or protein–nucleic acid interactions differ. *CsaRL1a*, *CsaRL3b*-*c*, and *CsaRL4a*-*c* have intron/exon structures that are comparable, although *CsaRL1b* and *CsaRL3a* each have a continuous open reading frame. The organization of *CsaRL1b* and *CsaRL3a* with a single exon is consistent with that of *Arabidopsis AtRLs*, which is most likely a derived state involving the loss of the second coding exon in the lineage [33]. The motif analysis revealed that motifs 1–3 are conserved across all CsaRLs, implying that these three motifs play essential roles in function regulation. The aberration in the C-terminal highlights the diverse ways CsaRLs contribute to the regulation of transcription. Sequence alignment revealed a very conserved SANT domain (comprising motif 1, motif 2, and a part of motif 3) at the N-terminal, which is required for CsaRLs to collaborate in complexes or bind to target genes in transcription and repression activities [40]. The SANT-MYB domain transcription factors regulate transcription by binding to sequence-specific DNA [41]. The CREB protein is essential in regulating the expression of cAMP. The activated CREB proteins bind to CREB-binding proteins (CBP), and CBP proteins participate in chromatin remodeling via histone acetyltransferase activity [11,42,43]. However, the “putative incomplete CREB 2 domain” in CsaRL1a, CsaRL3a, CsaRL3b, and CsaRL3c are substantially more incomplete than in CsaRL1b, CsaRL4a, CsaRL4b, CsaRL4c, AtRL2, and AmRAD (Figure 1C), which might suggest that CsaRL proteins participate in different function. The analysis demonstrated that the functional diversity among the CsaRLs in tea might be related to differences in gene architecture, protein properties, protein motifs, and C-terminal domains.

### 3.2. Evolution and Functional Annotation of the CsaRL Genes

Phylogenetic analysis can offer insights into plant lineage, evolutionary mechanisms, and protein function. RL proteins from many woody plants, such as *Populus*, *Malus domestica*, *Prunus persica*, and *Eucalyptus grandis*, were classified into Groups I, III, and V. The phylogenetic tree revealed that CsaRLs, on the other hand, were restricted to Group V. This suggested that CsaRLs and CssRLs have tighter genetic and paralogy links, which is consistent with the results from percent identity analysis. The presence of 158 RL proteins was found only in plants, ranging from algae to monocots and dicots. It implies that the RL duplication event occurred concurrently with plant sexual reproduction and occurred before the diversification of *Lamiales*, *lamiids*, or *asteroids* [44]. The segmental duplications of the *Arabidopsis* genome are assumed to have resulted from an ancient polyploidization event [34]. The RL family genes in *Yunkang-10* and *Shuchazao* had higher homology within varieties than between them, implying that segmental duplications in tea plants remained intact following the diploidization event. Furthermore, the computed Ka/Ks ratios of all *RL* gene pairs were less than one, indicating that purifying selection occurred during the evolution of RL genes in tea plants.

The evolutionary distribution of CsaRLs, as revealed by the phylogenetic tree, was closely correlated with the position of protein sequence motifs. This distribution mirrored protein homology within the same clade, allowing functional annotations to be created [45]. The over-expression of *AtRL2* in *Arabidopsis* resulted in the absence of apical hooks with short hypocotyls [16], the repression of floral transition (*Flowering Locus T*) [17], the modification of seed germination and seedling development [21], and the accumulation of chlorophyll [46]. The RL protein FSM1 is an early fruit-specific transcription factor in tomatoes that controls cell expansion by forming the FSM1/FSB1/MYBI complex [15]. CsaRL1a and CsaRL1b were homologous to AtRL1, AtRL2, and SlFSM1, indicating their comparable roles in tissue development, morphogenesis, and reproductivity. CsaRL3b had close homology with PtRAD1, implying that it, too, may play a role in xylem development [30]. Furthermore, we know that AtRL3 (XP_002869031.1) and AtRL4 (NP_001077912.1) are expressed in the rosette stem and leaf traces just outside the vascular bundles [33] and that OsRL3 (XP_015627089.1) mediates ABA-induced leaf senescence and salt sensitivity in rice [13]. Nonetheless, the phylogenetic tree shows that CsaRL proteins are very far from AtRL3, AtRL4, and OsRL3, implying functional differences between CsaRLs and AtRL3, AtRL4, and OsRL3.

### 3.3. CsaRL Is Likely to Play Essential Roles during Tissue Development

The tissue profiling of genes invariably reflects the partial molecular functions of related proteins. Many examples demonstrate this relationship between localization and function. The β-tubulin genes are differentially expressed in the stem and hypocotyl and play critical roles during fiber differentiation [47]. Flavonoid-, caffeine-, and theanine-biosynthesis-related genes are highly expressed in developing leaves [48,49]. The expression of GbRL2 in *Gossypium barbadense* L. was increased at elongating fibers and GbRL2 was supposed to be a target for genetic improvement of cotton fiber [50]. *AtRL1* and *AtRL2* are *Arabidopsis* genes expressed in the cotyledon, fruit septum, fruit valve, leaf, and hypocotyl, and have been linked to morphogenesis and reproduction [17,21]. PtRAD1 is upregulated in the xylem of *Populus* and affects xylem development [30]. In this study, *CsaRL1a* expression was minimal in flowers, buds, and upper stems, but significantly higher in tissues from the first-to-mature leaves, implying that it may be required for leaf development and leaf-specific metabolism rather than reproduction and meristem. The expression of *CsaRL1b* was the lowest in almost all tissues, despite being homologous to CsaRL1a. This could be because small RNAs (21–26 nucleotides long) suppress the expression of sequence homologous genes at the transcriptional level [51,52]. *CsaRL1b*, *CsaRL3a*, *CsaRL3b*, and *CsaRL3c* are expressed more in lower stems than in other tissues, implying that these genes are involved in xylem development. *CsaRL4c* was the most abundant gene in roots, indicating that it might play a key role in nutrition acquisition. Notably, the molecular functions of CsaRL1a and CsaRL3b determined by tissue profiling were consistent with the phylogenetic-based functional annotation, emphasizing the value of evolutionary research.

### 3.4. Potential Roles of CsaRL Genes in Response to Different Abiotic Stresses in Tea

Abiotic stressors, such as drought, high salinity, and extreme temperatures significantly impair plant growth, development, and metabolism. The phytohormone ABA and jasmonate participates in the abiotic stress response in plants and plays a vital role in integrating various stress signals [53,54]. The transcriptional regulation of stress-responsive genes is essential for improving plant abiotic stress tolerance. Previous research linked RL genes to several abiotic stress responses, including salinity and ABA treatment [13,21]. In this study, almost all *CsaRL* genes showed differential expression in response to drought, salt stress, and MeJA and ABA treatment. However, the expression pattern for all eight genes varied, indicating that *CsaRLs* may have different mechanisms for enhancing plant tolerance. At 48 h, *CsaRL1a*-*c*, *CsaRL3a*-*c*, and *CsaRL4a*-*c* were upregulated in response to PEG-6000, NaCl, MeJA, and ABA, indicating their positive roles in mitigating treatment. Our findings indicated that *CsaRL* genes participated in various abiotic stress responses, similar to *OsRL3* in rice [13] and *AtRL2* in Arabidopsis [21]. *CsaRL1a*, in particular, was expressed substantially more than other genes, with the highest increase under ABA treatment. In summary, characterizing *CsaRL* genes in response to environmental challenges would improve our understanding of functions in plant tolerance enhancement.

### 3.5. Potential Regulatory Roles of CsaRL1a in Tea

*CsaRL* expression profiles revealed that *CsaRL1a* was the most targeted in abundance, tissue specificity, and stress responses. Previous research has shown that the RL protein can interact with other proteins and create a complex to perform regulatory functions, similar to the RAD-DRIF complex in *Antirrhinum* [55], the FSM1-FSB1 complex in tomato [15], and the RAD1-DRIF1-WOX complex in *Populus* [30]. It may also regulate target genes by binding to the promoter, similar to how *AtRL2* regulates *ABI5* [21]. To investigate the regulatory role of *CsaRL1a*, we generated functional interaction networks of CsaRL1a using transcriptome data and visualized the networks using Lianchuan Omicstudio. CsaRL1a correlated favorably with stress-responsive proteins like WIN1-like and beta-amylase, as well as metabolism-related proteins like UFGT3-like and GDSL esterase/lipase. According to previous studies, ABA, cold, salt, and drought stress significantly affect WIN1-like [56] and beta-amylase [57]. UFGT3 is involved in anthocyanin accumulation [58], while GDSL esterases/lipases are essential to many physiological and biochemical processes, such as plant growth and development, organ morphogenesis, adversity stress, and lipid metabolism [59]. The networks partially explained the responses of CsaRL1a to environmental stress and predicted the regulatory function of CsaRL1a over primary and secondary metabolites in tea. Meanwhile, observing the growth and growth status of yeast containing the pGBKT7-CsaRLs vector on a nutrient-deficient medium (Figure 5) showed that CsaRLs did not possess transcriptional self-activating activity, suggesting CsaRLs are likely to perform their regulatory functions by interacting with other proteins.

## 4. Materials and Methods

### 4.1. Plant Material, Growing Conditions, and Abiotic Stress Treatments

Four-year-old *C. sinensis* var. *assamica* cv. *Yunkang-10* tea plants were planted in the tea garden of Guizhou University. Flower, bud, first leaf, second leaf, third leaf, fourth leaf, fifth leaf, upper stem, middle stem, and lower stem were among the 11 tissues utilized in the qRT-PCR experiment. *Yunkang-10* seedlings were sown in plastic containers with peat soil for the abiotic stress experiment. We moved the pots inside the artificial climate chamber (Lisk Instrument, Nanjing, China) once the seeds germinated. The seedlings were cultivated for one month under controlled conditions of 16 h light/8 h dark per day, with a temperature of 25 ± 2 °C and a relative humidity of 70%. When the seedlings reached the one bud and two leaves stage, the seedlings with consistent growth were divided into five groups with 16 members each and transplanted into peat soil. The new settings were 25 °C/16 h, 20 °C/8 h photoperiod, 80% air humidity, and 200 µmol light intensity. After one week of acclimatization, stress treatments were imposed using 25% PEG-6000 (Shanghai Aladdin Biochemical Technology Co., Ltd., Shanghai, China), 200 mmol/L NaCl (Sangon Biotech, Shanghai, China), 200 µmol/L ABA (Sangon Biotech, Shanghai, China), and 1 mmol/L MeJA (Nanjing Warbio, Nanjing, China) solution. The experimental sample was a combination of first and second leaves after 0, 12, 24, 48, and 72 h of treatment. The samples were frozen in liquid nitrogen and stored at −80 °C.

### 4.2. Identifying and Cloning the Radialis-like Family Genes in the C. sinensis var. assamica cv. Yunkang-10

The *Arabidopsis*, *Oryza sativa*, and *Solanum lycopersicum* RL proteins were downloaded and utilized as queries against the local protein database using BLAST-2.7.1+ (E-value < 1 × 10^−5^) for *CsaRL* identification. The hidden Markov model (HMM) of the SANT domain (PF00249) was created using the Pfam database (https://pfam.xfam.org/, accessed on 23 October 2022) [60]. Following the retrieval of the putative CsaRLs, the SANT domain was validated using SMART (http://smart.embl-heidelberg.de/, accessed on 23 October 2022), InterProScan (http://www.ebi.ac.uk/interpro/search/sequence/, accessed on 23 October 2022) [61,62], and BLASTP (https://blast.ncbi.nlm.nih.gov/Blast.cgi, accessed on 23 October 2022). Sequence-specific primers for *CsaRL* genes were designed using SnapGene Viewer5.2.4 (Appendix A). The *CsaRL* family genes were cloned using RT-PCR and then sequenced to confirm identity by the Sangon Biotech Company (Shanghai, China). The percent identity between *CsaRLs* and *CssRLs* was computed through BLASTN (https://blast.ncbi.nlm.nih.gov/Blast.cgi, accessed on 5 November 2022). Gene structure analysis was performed using TBtools software (TBtools-II v1.108) [63], while translation into protein sequences was achieved using GENETYX software (GENETYX Ver.14). ExPASy (https://web.expasy.org/protparam/, accessed on 5 November 2022) [64] calculated the physical and chemical parameters of each CsaRL protein, including the molecular weight, isoelectric point (p*I*), aliphatic index, and grand average of hydropathicity (GRAVY). Subcellular location was predicted using Plant-mPLoc (http://www.csbio.sjtu.edu.cn/bioinf/plant-multi/, accessed on 10 November 2022) and protein motifs were discovered by using MEME (https://meme-suite.org/meme/, accessed on 10 November 2022) [65]. Ka (nonsynonymous substitution rate) and Ks (synonymous substitution rate), as well as the evolution constraint (Ka/Ks) between paralogous pairs of *RL* genes in *Yunkang-10* and Shuchazao, were computed using TBtools (TBtools-II v1.108) [66].

### 4.3. Phylogenetic Analysis

158 RL proteins from 18 plant species were downloaded from tea genome data sites (http://www.plantkindomgdomgdb.com/tea_tree/, accessed on 12 November 2022), the National Center for Biotechnology Information (NCBI) protein database (https://www.ncbi.nlm.nih.gov/, accessed on 12 November 2022), and Phytozome (http://phytozome-next.jgi.doe.gov, accessed on 12 November 2022). The protein Csa022968.1 (transcription factor DIVARICATA) was chosen as the outgroup. MEGA7 [66] was used to perform multiple sequence alignments and build a phylogenetic tree using the maximum likelihood (ML) method (bootstrap option n = 1000). The Newick tree was submitted to Interactive Tree of Life (https://itol.embl.de/, accessed on 15 November 2022) [67] for displaying, manipulating, and annotating phylogenetic trees.

### 4.4. RNA Isolation and Quantitative RT-PCR

Following the manufacturer’s instructions, total RNA was extracted using an RNA extraction kit (Omega Bio-Tek, Shanghai, China) and reverse transcribed into cDNA using the EasyScript^®^ One-Step gDNA Removal and cDNA Synthesis SuperMix (Transgen, Beijing, China). The qRT-PCR was carried out on a Bio-Rad CFX platform (Bio-Rad, Hercules, CA, USA) with TransStart^®^ Top Green qPCR SuperMix (Transgen, Beijing, China) using the following program: 95 °C for 30 s, then 40 cycles at 95 °C for 10 s and 60 °C for 30 s, followed by melt curves at 65 °C for 10 s and 0.5 °C increments to 95 °C. We selected the reference β-actin gene as an internal standard for normalization [22]. Appendix A contains a list of the primers used in the qRT-PCR.

### 4.5. Transcriptional Activation Analysis of the Radialis-like Family Genes in the C. sinensis var. assamica cv. Yunkang-10

A yeast assay was used to perform the transcriptional activity experiment. CsaRLs were amplified using specialized primers incorporating EcoR I and BamH I recognition sites (Appendix A). The coding sequence of CsaRLs was cloned into the pGBKT7 vector to create the BD-CsaRLs expression vector. The BD-CsaRLs vector and a control BD empty vector were transformed into *Saccharomyces cerevisiae* Y2H Gold (Weidi Biotechnology Co., Ltd., Shanghai, China). We used the yeast expression vector pGBKT7 as the negative control and the plasmid pGBKT7-GAL4 as the positive control. The yeast cells were grown for three days at 30 °C in a synthetic defined (SD) medium (Sangon Biotech Co., Ltd., Shanghai, China) lacking tryptophan (SD/-Trp). For 3–5 days, the transformed yeast cells were grown in SD media lacking tryptophan, histidine, and adenine (SD/-Trp/-His/-Ade/X-a-gal). CsaRLs transcriptional activity was measured using an X-α-galactosidase assay (Sangon Biotech Co., Ltd., Shanghai, China)

### 4.6. Functional Interaction Networks of Proteins

The transcriptome dataset of *Yunkang-10* subjected to drought stress (NCBI SRA accession: PRJNA564414) was used to compute Pearson’s correlation of CsaRL1a and its target proteins, with rho > 0.9 and rho < −0.9 [22]. These data were uploaded to Lianchuan Omicstudio (https://www.omicstudio.cn/index, accessed on 25 November 2022) to visualize the CsaRL1a protein–protein interaction network.

### 4.7. Statistical Analysis

Relative transcript abundance was calculated using the comparative 2^−ΔΔCT^ method [68]. Using the SPSS statistics software (IBM SPSS statistics 26), a two-tailed Student’s *t*-test was used to determine the statistical significance and the significant differences between groups. A *p* value of < 0.05 was considered statistically significant. The results are the mean values ± SE of three biological replicates. The gene expression was normalized and plotted in separate heatmaps in different tissues and under various abiotic conditions. Advanced heatmap plots were performed using the OmicStudio tools at https://www.omicstudio.cn (accessed on 12 February 2023).

## 5. Conclusions

We identified and cloned eight *CsaRL* genes using information from the genome of *C. sinensis* var. *assamica* cv. *Yunkang-10* and analyzed them for gene structure, protein motif, conserved domains, protein properties, and phylogenetic analysis. Tissue-specific profiling suggested that *CsaRLs* are most likely involved in tissue development and metabolite biosynthesis. Expression patterns under various abiotic stress conditions demonstrated that *CsaRLs* exert regulatory functions over plant tolerance. *CsaRL1a* had distinct characteristics related to expression abundance, tissue specificity, and stress responses. None of the CsaRL proteins possess transcriptional activation ability in yeast cells. The protein–protein interaction network revealed that CsaRL1a might strongly correlate with stress response and metabolic biosynthesis, which is congruent with the discovery made by the qRT-PCR experiment. Although the research on the molecular function of the *CsaRL* genes is relatively rare, this study provides some data support for further research into their potential functions in regulating developmental processes, metabolite biosynthesis, and stress response in the tea plant.

## Figures and Tables

**Figure 1 plants-12-03039-f001:**
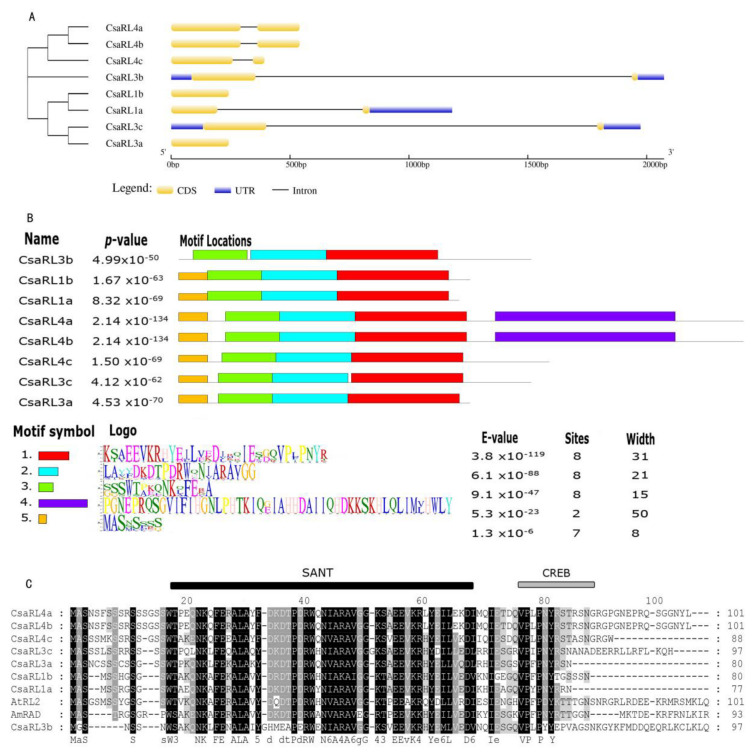
The exon-intron structures, motifs, and analysis of the conserved domain sequence of *CsaRLs*. (**A**) Exon-intron structure of *CsaRL* genes. Blue boxes represent untranslated upstream/downstream regions; yellow boxes represent exons; and lines indicate introns. (**B**) Conserved motifs of CsaRL proteins. Eight conserved motifs of CsaRL proteins were identified using the MEME online tool. Each motif in the CsaRL proteins was represented by a different color. (**C**) Multiple alignments of amino acid sequences of the Radialis family genes in the *C. sinensis* var. *assamica* cv. *Yunkang-10*, and AtRL2 and AmRAD proteins. The black line indicates the conserved DNA-binding domain.

**Figure 2 plants-12-03039-f002:**
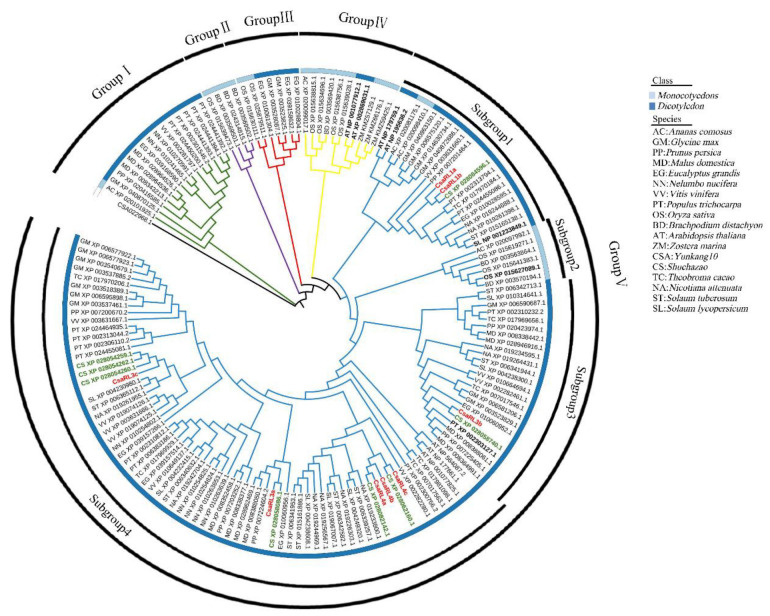
Phylogenetic analysis of RADIALIS-like (RL) proteins from eighteen distinct species. A total of 158 RL protein sequences were analyzed to construct a phylogenetic tree using the neighbor-joining method with 1000 repeated bootstrap tests, distance, and pairwise deletion. The branches with green, violet, red, yellow, and blue colors were classified into Group I, Group II, Group III, Group IV, and Group V, respectively.

**Figure 3 plants-12-03039-f003:**
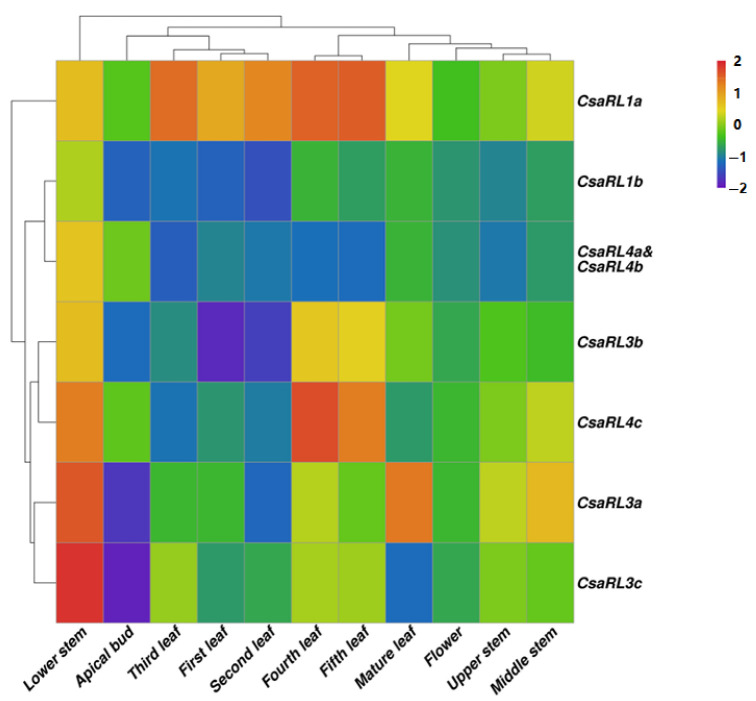
Expression profiles of *CsaRLs* in different tissues of the tea plant. qRT-PCR was used to measure the *CsaRL* transcript abundance, and the results were calculated using the 2^−ΔΔCt^ method. The color bar in the top right-hand corner of the heat map indicates higher expression levels in red and lower expression levels in blue.

**Figure 4 plants-12-03039-f004:**
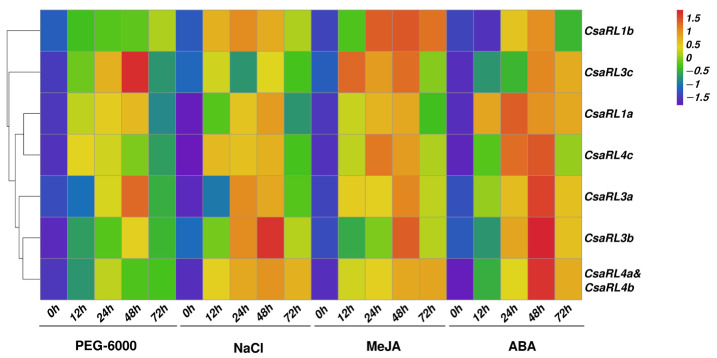
The expression patterns of *CsaRL* genes under different abiotic stresses in tea plants. The relative expression levels of each *CsaRL* gene were evaluated using qRT-PCR and the 2^−ΔΔCt^ method following treatment with PEG-6000, NaCl, MeJA, and ABA. The 0 h samples were designated as controls. The log 2 values of the calculated results generated the heat map. The intensity value bar displayed on the left side has red and green colors representing higher and lower expression values, respectively, compared to the control. The value for each time point represents the mean of three biological replicates.

**Figure 5 plants-12-03039-f005:**
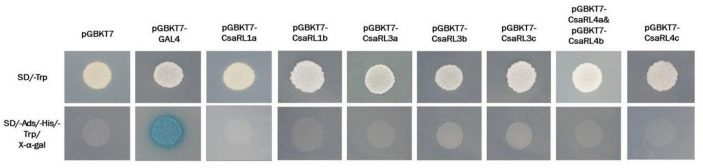
Transcriptional activation of CsaRLs in *Saccharomyces cerevisiae*. *CsaRLs* were subcloned into a pGBKT7 vector and transformed into yeast cells. The transformed yeast cells were cultured on selective SD medium (SD/-Trp) and then selected using X-α-Gal (SD/-Trp/-His/-Ade) assays. pGBKT7-GAL4: positive control; pGBKT7: negative control.

**Figure 6 plants-12-03039-f006:**
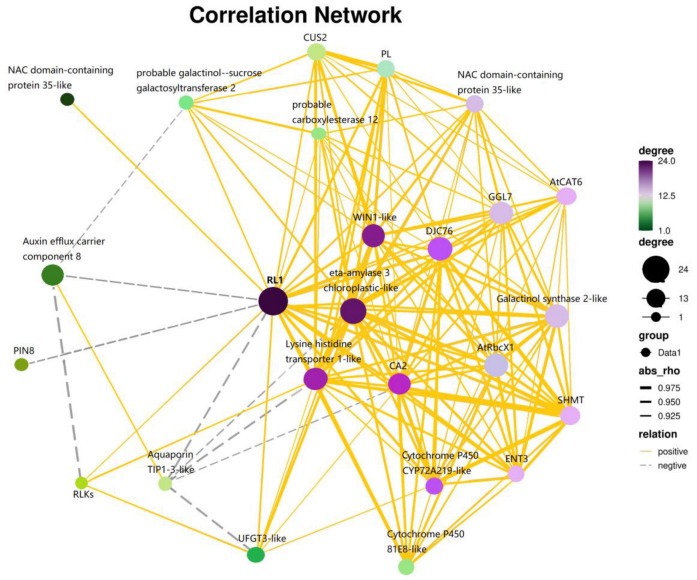
Protein–protein interaction networks of CsaRL1a. Nodes represent proteins, while yellow lines indicate interactions between proteins in the hormone-signaling pathway.

**Table 1 plants-12-03039-t001:** Eight *CsaRL* family genes identified in *Yunkang-10*.

Gene Name	Gene ID of CSA	Chromosome Location	Cloned CDS (bp)	Description	Gene ID of CSS	Per. Ident
*CsaRL4a*	CSA010734.1	Sc0004192	306	*CssRL4*	XM_028226341.1	99.67%
*CsaRL4b*	CSA017368.1	xfSc0000195	306	*CssRL4*	XM_028226341.1	99.67%
*CsaRL1b*	CSA011543.1	xpSc0053403	243	*CssRL1*	XM_028228795.1	99.56%
*CsaRL3a*	CSA018425.1	Sc0000237	243	*CssRL3*	XM_028202285.1	99.59%
*CsaRL4c*	CSA017695.1	Sc0000652	267	*CssRL4-X2*	XM_028206360.1	99.25%
*CsaRL1a*	CSA027066.1	Sc0000099	234	*CssRL1*	XM_028228795.1	89.24%
*CsaRL3b*	CSA032285.1	Sc0000093	294	*CssRL3*	XM_028202939.1	98.98%
*CsaRL3c*	CSA011662.1	Sc0001335	294	*CssRL3*	XM_028230429.1	98.98%

**Table 2 plants-12-03039-t002:** Bioinformatic analysis of CsaRL proteins. MW–Molecular weight; p*I*— theoretical isoelectric point protein.

Name	Cloned Protein (aa)	MW (kDa)	p*I*	Aliphatic Index	GRAVY	Localization
CsaRL4a	101	11.38	8.01	49.31	−1.111	Nucleus.
CsaRL4b	101	11.38	8.01	49.31	−1.111	Nucleus.
CsaRL1a	77	8.83	8.03	60.91	−0.892	Nucleus.
CsaRL3a	80	8.99	9.03	54.88	−0.907	Nucleus.
CsaRL4c	88	9.94	8.01	58.75	−0.873	Nucleus.
CsaRL1b	80	8.81	6.83	57.38	−0.775	Nucleus.
CsaRL3b	97	11.11	5.67	68.35	−0.777	Nucleus.
CsaRL3c	97	11.23	9.76	73.51	−0.972	Nucleus.

## Data Availability

Publicly available datasets were analyzed in this study. This data can be found here: [NCBI SRA accession: PRJNA564414].

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
