# Peer review of "Genome-Wide Identification and Expression Analysis of the RADIALIS-like Gene Family in Camellia sinensis"

_plants, 2023, doi:10.3390/plants12173039_

Round 1

Reviewer 1 Report (New Reviewer)

The introduction provided sufficient background and included all relevant references. Research method was good, and the methods were well described. Figures were clear with sufficient visualization of the results. The only recommendation I have is to mention the full format for any abbreviations you used at the first mention.

MYB, ABA, SANT and etc.

Author Response

Dear reviewers

Re: Manuscript ID: plants-2486492 and Title: Genome-wide Identification and Expression Analysis of the RADIALIS-like Gene Family in Camellia sinensis

Thank you for your letter and the reviewers’ comments concerning our manuscript entitled “Genome-wide Identification and Expression Analysis of the RADIALIS-like Gene Family in Camellia sinensis” (ID). Those comments are valuable and very helpful. We have read through comments carefully and have made corrections. Based on the instructions provided in your letter, we uploaded the file of the revised manuscript. The responses to the reviewer's comments are marked in red and presented following. We would love to thank you for allowing us to resubmit a revised copy of the manuscript and we highly appreciate your time and consideration.

Sincerely.

Q1. The only recommendation I have is to mention the full format for any abbreviations you used at the first mention.

Response:We apologize for the abbreviations problems in the original manuscript. We have read the entire text again and improved the issue of all abbreviations.

The improvements are as follows.v-myb avian myeloblastosis viral oncogene homolog (MYB)

SANT Associated domain (SANT)

Abscisic acid (ABA)

Reviewer 2 Report (New Reviewer)

The abstract needs to be improved including main sections of methodology. Currently the methodology is lacking in the manuscript.

Author Response

Dear reviewers

Re: Manuscript ID: plants-2486492 and Title: Genome-wide Identification and Expression Analysis of the RADIALIS-like Gene Family in Camellia sinensis

Thank you for your letter and the reviewers’ comments concerning our manuscript entitled “Genome-wide Identification and Expression Analysis of the RADIALIS-like Gene Family in Camellia sinensis” (ID). Those comments are valuable and very helpful. We have read through comments carefully and have made corrections. Based on the instructions provided in your letter, we uploaded the file of the revised manuscript. The responses to the reviewer's comments are marked in red and presented following. We would love to thank you for allowing us to resubmit a revised copy of the manuscript and we highly appreciate your time and consideration.

Sincerely.

Q1. The abstract needs to be improved including main sections of methodology. Currently the methodology is lacking in the manuscript.

Response:We apologize for the abstract problems in the original manuscript. We have made modifications to the abstract section of the article and added methods. The details are as follows.

Abstract: The RADIALIS-like (RL) proteins are v-myb avian myeloblastosis viral oncogene homolog (MYB)-related transcription factors (TFs) involved in many biological processes, including metabolism, development, and response to biotic and abiotic stresses. However, the studies on RL genes of Camellia sinensis are not comprehensive enough. Therefore, we undertook this study and identified 8 CsaRLs based on the typical conserved domain SANT Associated domain (SANT) of RL. These genes have low molecular weights and theoretical pI values ranging from 5.67 to 9.76. Gene structure analysis revealed that six CsaRL genes comprise two exons and one intron, while the other two contain a single exon encompassing motifs 1, 2, and part of motif 3. The phylogenetic analysis divided one hundred and fifty-eight RL proteins into five primary classes, in whichCsaRLs clustered in Group V and were homologous with CssRLs of the Shuchazao variety. In addition, we selected different tissue parts to analyze the expression profile of CsaRLs, and the results show that almost all genes displayed variable expression levels across tissues, with CsaRL1a relatively abundant in all tissues. qRT-PCR (real-time fluorescence quantitative PCR) was used to detect the relative expression levels of the CsaRL genes under various abiotic stimuli, and it was found that CsaRL1a expression levels were substantially higher than other genes, with Abscisic acid (ABA) causing the highest expression. The self-activation assay with yeast two-hybrid system showed that CsaRL1a has no transcriptional activity. According to protein functional interaction networks, CsaRL1a was well connected with WIN1-like, lysine histidine transporter-1-like, β-amylase 3 chloroplastic-like, carbonic anhydrase-2-like (CA2), and carbonic anhydrase dnaJC76 (DJC76). This study adds to our understanding of the RL family and lays the groundwork for further research into the function and regulatory mechanisms of the CsaRLs gene family in Camellia sinensis.

Reviewer 3 Report (New Reviewer)

In this study, Yang et al. conducted an investigation to identify RL proteins in Camellia sinensis and conducted an analysis of the expression levels of these CasRLs in various tissues and under abiotic stresses. The organization of the study is coherent and clear. This research significantly contributes to our understanding of the roles of the RL gene family in Camellia sinensis. However, to enhance the quality of this manuscript, several concerns need to be addressed:

(1) The manuscript lacks proper citations for the bioinformatics tools used in this study. It is essential to provide references for these tools, allowing readers to access further information and verify the methodologies used.

(2) More comprehensive details regarding the data analysis should be included. This entails specifying the versions of the tools used and providing the running parameters associated with each software. Such information is crucial for reproducibility and to enable others to validate the findings.

(3) If data from other studies, such as the transcriptome data mentioned on Page 9 Line 263, are utilized in this research, appropriate citations or references should be provided to acknowledge the sources. This will ensure proper credit to the original researchers and allow readers to explore the relevant data for better contextual understanding.

By addressing these concerns, the manuscript will be significantly improved and better suited for publication. The revision will not only enhance the overall quality of the study but also provide readers with more confidence in the findings and conclusions presented.

The writing throughout the manuscript requires further improvement and polishing. This includes addressing issues related to grammar, syntax, and sentence structure to ensure clarity and coherence.

Author Response

Dear reviewers

Re: Manuscript ID: plants-2486492 and Title: Genome-wide Identification and Expression Analysis of the RADIALIS-like Gene Family in Camellia sinensis

Thank you for your letter and the reviewers’ comments concerning our manuscript entitled “Genome-wide Identification and Expression Analysis of the RADIALIS-like Gene Family in Camellia sinensis” (ID). Those comments are valuable and very helpful. We have read through comments carefully and have made corrections. Based on the instructions provided in your letter, we uploaded the file of the revised manuscript. The responses to the reviewer's comments are marked in red and presented following. We would love to thank you for allowing us to resubmit a revised copy of the manuscript and we highly appreciate your time and consideration.

Sincerely.

Q1. (1) The manuscript lacks proper citations for the bioinformatics tools used in this study. It is essential to provide references for these tools, allowing readers to access further information and verify the methodologies used.

Response:We have added references 63-71 for bioinformatics tools such as Pfam database, InterProScan, ExPASy, etc. to the article

Q2. (2) More comprehensive details regarding the data analysis should be included. This entails specifying the versions of the tools used and providing the running parameters associated with each software. Such information is crucial for reproducibility and to enable others to validate the findings.

Response:We have provided specific versions of software such as TBtools and SPSS in the article, and also provided details on the application of data analysis software SPSS in the article

Q3. (3) If data from other studies, such as the transcriptome data mentioned on Page 9 Line 263, are utilized in this research, appropriate citations or references should be provided to acknowledge the sources. This will ensure proper credit to the original researchers and allow readers to explore the relevant data for better contextual understanding.

Response:The transcriptome data used in the article is provided on page 15, line 509, with a reference number for NCBI (NCBI SRA access: PRJNA564414). The article also cites the author's published article

This manuscript is a resubmission of an earlier submission. The following is a list of the peer review reports and author responses from that submission.

Round 1

Reviewer 1 Report

There are many mistakes in the writing process of the manuscript, I hope the author can correct it;and it is best to ask an expert to revise the manuscript as a whole to improve the quality of English;

Some abbreviations appear in the paper, when they first appear, need a full name;

Figure 1, The font type and size are very inconsistent, and it is recommended that the author unify the font in all figures;

Latin should be italicized;

There are too many reference questions, it is recommended to find an up-to-date article in Plants and refer to its format.

There are many mistakes in the writing process of the manuscript, I hope the author can correct it;and it is best to ask an expert to revise the manuscript as a whole to improve the quality of English;

Some abbreviations appear in the paper, when they first appear, need a full name;

Figure 1, The font type and size are very inconsistent, and it is recommended that the author unify the font in all figures;

Latin should be italicized;

There are too many reference questions, it is recommended to find an up-to-date article in Plants and refer to its format.

There are many mistakes in the writing process of the manuscript, I hope the author can correct it;and it is best to ask an expert to revise the manuscript as a whole to improve the quality of English;

Some abbreviations appear in the paper, when they first appear, need a full name;

Figure 1, The font type and size are very inconsistent, and it is recommended that the author unify the font in all figures;

Latin should be italicized;

There are too many reference questions, it is recommended to find an up-to-date article in Plants and refer to its format.

There are many mistakes in the writing process of the manuscript, I hope the author can correct it;and it is best to ask an expert to revise the manuscript as a whole to improve the quality of English;

Some abbreviations appear in the paper, when they first appear, need a full name;

Figure 1, The font type and size are very inconsistent, and it is recommended that the author unify the font in all figures;

Latin should be italicized;

There are too many reference questions, it is recommended to find an up-to-date article in Plants and refer to its format.

There are many mistakes in the writing process of the manuscript, I hope the author can correct it;and it is best to ask an expert to revise the manuscript as a whole to improve the quality of English;

Some abbreviations appear in the paper, when they first appear, need a full name;

Figure 1, The font type and size are very inconsistent, and it is recommended that the author unify the font in all figures;

Latin should be italicized;

There are too many reference questions, it is recommended to find an up-to-date article in Plants and refer to its format.

There are many mistakes in the writing process of the manuscript, I hope the author can correct it;and it is best to ask an expert to revise the manuscript as a whole to improve the quality of English;

Some abbreviations appear in the paper, when they first appear, need a full name;

Figure 1, The font type and size are very inconsistent, and it is recommended that the author unify the font in all figures;

Latin should be italicized;

There are too many reference questions, it is recommended to find an up-to-date article in Plants and refer to its format.

There are many mistakes in the writing process of the manuscript, I hope the author can correct it;and it is best to ask an expert to revise the manuscript as a whole to improve the quality of English;

Some abbreviations appear in the paper, when they first appear, need a full name;

Figure 1, The font type and size are very inconsistent, and it is recommended that the author unify the font in all figures;

Latin should be italicized;

There are too many reference questions, it is recommended to find an up-to-date article in Plants and refer to its format.

There are many mistakes in the writing process of the manuscript, I hope the author can correct it;and it is best to ask an expert to revise the manuscript as a whole to improve the quality of English;

Some abbreviations appear in the paper, when they first appear, need a full name;

Figure 1, The font type and size are very inconsistent, and it is recommended that the author unify the font in all figures;

Latin should be italicized;

There are too many reference questions, it is recommended to find an up-to-date article in Plants and refer to its format.

Author Response

Thank you for your suggestions and comments. We apologize for the incomplete manuscript. After some time of revision, we have made every effort to improve it as much as possible. Now I will respond to your suggestions one by one.

  1. There are many mistakes in the writing process of the manuscript, I hope the author can correct it;and it is best to ask an expert to revise the manuscript as a whole to improve the quality of English

Response: The article has been polished by "editeg" company(languagehttps://www.editeg.com/) to enhance the professional level of language.

  1. Some abbreviations appear in the paper, when they first appear, need a full name

Response: The abbreviations that first appeared in the article have all been supplemented with their full names

  1. Figure 1, The font type and size are very inconsistent, and it is recommended that the author unify the font in all figures

Response: The text size and font of the images in the article are all consistent. Thank you for your valuable suggestion

  1. Latin should be italicized;There are too many reference questions, it is recommended to find an up-to-date article in Plants and refer to its format

Response: Latin has been modified to italicize. We have found the latest PLANTS article to modify the format of the reference materials for reference. Thank you for your suggestion

Reviewer 2 Report

The manuscript "Genome-wide Identification and Expression Analysis of the RADIALIS-like Gene Family in Camellia Sinensis" by Yang et al. concerns the characterization of a family of trancription factors in C. sinensis, particularly their molecular characteristics and their response to abiotic stress. The authors develop several in silico and in vivo studies to characterize these genes.

However, the work is far from ready for publication. Many mistakes can be found in the text, including grammar and vocabulary. Moreover, there are several non-sense sentences and typos throughout the manuscript, as well as obvious statements that do not provide anything valuable to the text. Besides, authors mix sections, putting information about materials and methods in the wrong sections, as well as with results within the discussion section. Authors claim there is one Supplementary file but I found three, some of them in Chinese, which at this moment I am unable to understand. I will not go into details, the whole manuscript needs to be rewritten.

Other comments:

- Heatmaps showing what I understand statistical differences with letters are kind of new to me, there are better ways to show these sort of data.

- The authors talk about cultivar Shuchazao, but it is not clear where that data is or who analyzed it.

- Lines 342-343: certainly algae are no flowering plants.

Author Response

Thank you for your suggestions and comments. We apologize for the incomplete manuscript. After some time of revision, we have made every effort to improve it as much as possible. Now I will respond to your suggestions one by one.

1.The work is far from ready for publication. Many mistakes can be found in the text, including grammar and vocabulary.

Response: The article has been polished by "editeg" company(languagehttps://www.editeg.com/) to enhance the professional level of language

2.There are several non-sense sentences and typos throughout the manuscript

Response: The meaningless sentences and typos in the article have been removed

3.Authors mix sections, putting information about materials and methods in the wrong sections, as well as with results within the discussion section

Response: Please refer to lines 424 and 440. We have placed the corresponding sections in the correct ones. Thank you for your valuable suggestion

  1. Authors claim there is one Supplementary file but I found three, some of them in Chinese, which at this moment I am unable to understand. I will not go into details, the whole manuscript needs to be rewritten

Response: We are sorry that the supplementary documents were missing from the previous submission, and this manuscript has been supplemented. The manuscript has been rewritten. Thank you for your suggestion

5.Heatmaps showing what I understand statistical differences with letters are kind of new to me, there are better ways to show these sort of data

Response: According to the supervisor's opinion, the heat map is not suitable for using letter statistics for differences, and letter statistics have been removed

6.The authors talk about cultivar Shuchazao, but it is not clear where that data is or who analyzed it

Response: There is relevant research information on Shucha Zao on this website(tpdb.shengxin.ren)

7.Lines 342-343: certainly algae are no flowering plants

Response: This sentence was written incorrectly and has been corrected. Thank you for your valuable advice

Round 2

Reviewer 1 Report

Please see the mauscript, which I have made some marks.

Reviewer 2 Report

None.